# Porous Venturi-Orifice Microbubble Generator for Oxygen Dissolution in Water

**Kelly Chung Shi Liew [1], Athina Rasdi [1], Wiratni Budhijanto [2,\*], Mohd Hizami Mohd Yusoff [1,3], Muhmmad Roil Bilad [1,3,\*], Norazanita Shamsuddin [4], Nik Abdul Hadi Md Nordin [1] and Zulfan Adi Putra [5]**

[1] Department of Chemical Engineering, Universiti Teknologi PETRONAS, Bandar Seri Iskandar, Perak 32610, Malaysia; kelly_19000174@utp.edu.my (K.C.S.L.); wannoorathina@gmail.com (A.R.); hizami.yusoff@utp.edu.my (M.H.M.Y.); nahadi.sapiaa@utp.edu.my (N.A.H.M.N.)

[2] Departemen Teknik Kimia, Fakultas Teknik, Universitas Gadjah Mada, Jl. Grafika No. 2, Kampus UGM, Yogyakarta 55281, Indonesia

[3] HICoE-Centre for Biofuel and Biochemical Research, Institute of Self-Sustainable Building, Universiti Teknologi PETRONAS, 32610, Seri Iskandar, Perak 32610, Malaysia

[4] Faculty of Integrated Technologies, Universiti Brunei Darussalam, Jalan Tungku Link, Gadong BE 1410, Brunei; norazanita.shamsudin@ubd.edu.bn

[5] PETRONAS Group Technical Solutions, Process Simulation and Optimization, Level 16, Tower 3, Kuala Lumpur Convention Center, Kuala Lumpur 50088, Malaysia; zulfan.adiputra@petronas.com.my

\* Correspondence: wiratni@ugm.ac.id (W.B.); mroil.bilad@utp.edu.my (M.R.B.); Tel.: +60-5-3658-7646 (M.R.B.)

**Abstract:** Microbubbles with slow rising speed, higher specific area and greater oxygen dissolution are desired to enhance gas/liquid mass transfer rate. Such attributes are very important to tackle challenges on the low efficiency of gas/liquid mass transfer that occurs in aerobic wastewater treatment systems or in the aquaculture industries. Many reports focus on the formation mechanisms of the microbubbles, but with less emphasis on the system optimization and assessment of the aeration efficiency. This work assesses the performance and evaluates the aeration efficiency of a porous venturi-orifice microbubble generator (MBG). The increment of stream velocity along the venturi pathway and orifice ring leads to a pressure drop ($P_{atm} > P_{abs}$) and subsequently to increased cavitation. The experiments were run under three conditions: various liquid velocity ($Q_L$) of 2.35–2.60 m/s at fixed gas velocity ($Q_g$) of 3 L/min; various $Q_g$ of 1–5 L/min at fixed $Q_L$ of 2.46 m/s; and free flowing air at variable $Q_L$s. Results show that increasing liquid velocities from 2.35 to 2.60 m/s imposes higher vacuum pressure of 0.84 to 2.27 kPa. They correspond to free-flowing air at rates of 3.2–5.6 L/min. When the system was tested at constant air velocity of 3 L/min and under variable liquid velocities, the oxygen dissolution rate peaks at liquid velocity of 2.46 m/s, which also provides the highest volumetric mass transfer coefficient ($K_La$) of 0.041 min$^{-1}$ and the highest aeration efficiency of 0.287 kgO$_2$/kWh. Under free-flowing air, the impact of $Q_L$ is significant at a range of 2.35 to 2.46 m/s until reaching a plateau $K_La$ value of 0.0416 min$^{-1}$. The pattern of the $K_La$ trend is mirrored by the aeration efficiency that reached the maximum value of 0.424 kgO$_2$/kWh. The findings on the aeration efficiency reveals that the venturi-orifice MBG can be further optimized by focusing on the trade-off between air bubble size and the air volumetric velocity to balance between the amount of available oxygen to be transferred and the rate of the oxygen transfer.

**Keywords:** microbubble generator; oxygen dissolution; aeration efficiency; venturi-orifice type; aeration efficiency

## 1. Introduction

Microbubble-based processes have emerged as a promising option for enhancing interphases mass-transfer for industrial applications [1]. The application of microbubbles in the aquaculture industry helps to enhance productivity, water quality, hydroponic plant growth and soil fermentation [2]. For example, microbubble generators (MBG) have been used in the farming of oyster [3,4] for promoting growth, shell opening and the increment of oyster's blood flow rate, ascertaining the beneficial effect on the bioactivity [3]. In intensive aquaculture of tilapia fish, the application of MGB, as an aerator, also promoted the growth rate of fish (both their length and weight) [5]. A special type of MBG in a form of bubble-jet-type air-lift pumps has also been applied for purifying fishery wastewater [6,7]. Recently, a membrane-based bubble generator has also been applied for cultivation of microalgae and aerobic wastewater treatment [8–10] and can potentially be used to enhance the efficiency for $CO_2$ dissolution for microalgae cultivation [11].

Microbubbles are generated through three fundamental methods: pressurization dissolution (decompression), rotating-flow (spiral flow) and cavitation for ejector and/or venturi methods [10,12,13]. These basic methods are the base for most of the recent modifications and optimizations [2]. Some of the recent developments include a system based on a porous media, constant flow nozzles and membrane or gas spargers coupled with a mixer (i.e., impeller) [12].

For the pressurized type MBG, the highly saturated gas is injected into the tank through a nozzle, together with the pressurized water to enhance gas dissolubility. The liquid-gas mixture then forms the microbubbles, due to a sudden pressure drop when flashed by a reducing valve at lower pressure [12,13]. The spiral flow liquid MBG is commonly designed in conical shape, to enhance the gas-water circulation. Water is introduced tangentially into the cylindrical tank to form a spiral pattern flow with a maelstrom-like cavity [4,6]. Meanwhile, spiral or swirl-based flow MBG can also work based on a self-suction mechanism for gas supply like orifice or venturi type MBG [14]. The gas is sucked in from the opening at the bottom of the tank towards the reduced pressure central core of the whirlpool. Then, the gas-liquid mixture is reduced into microbubbles, due to the shear effect of the centrifugation formed by the rapid rotating liquid flow [12,15].

The venturi effect has also been exploited to generate microbubbles; and the factors affecting microbubbles formation have widely been discussed. It consists of a converging-diverging nozzle with a throat at the middle [16,17]. When liquid enters the throat at a greater velocity, it lowers the static pressure, and this effect can be used for air suction and the subsequent formation of microbubbles (static pressure falls below the atmospheric pressure) [18]. Orifice type MBGs work under similar principles with the venturi type MBGs, in which the velocity change is also used as a decompressor [14,19]. Fujiwara et al. (2003) [20] investigated the phenomena of microbubble generation in a venturi tube with the use of 3-pentanol as surfactant. They found an inverse proportional relationship between pressure and velocity changes, and a directly proportional relationship between bubble diameter towards velocity along the venturi tube. The low local pressure within the venturi tube promotes cavitation generation conditions, but soon the formed void collapses and the pressure is recovered further in the downstream. However, Kaushik and Chel (2014) [18] reported an issue of immediate coalesce of microbubbles into bigger bubbles at the venturi discharge point, which can be limited by application of surfactant dosing. Fujiwara et al. (2003) [20] observed the bubbles formation and breakdown process under low liquid velocity ($Q_L$) of 4.2 L/min and high liquid velocity of 6.7 L/min. At lower liquid velocity, the bubble collapses gradually along the flow; while at higher liquid velocity, bubbles fission occurs at the front/top surface of a single large bubble, at a further section of the venturi tube [20]. The observations suggest that microbubbles formation could be based on two mechanisms: by the shearing motion [7] of liquid under lower liquid velocity, and by the sudden recovery of the pressure under higher liquid velocity [20].

The gas (naturally forming bubbles after being forced/sucked into liquid) and the cavitation effect contribute to microbubbles formation. Sadatomi and Kawahara (2008) proposed a concept of automated gas suction under negative pressure in the throat [21]. Ejector-type MBG that works based on cavitation

also falls under this category [22]. According to Terasaka et al. (2011), a typical ejector-type MBG refers to a liquid flow channel that involves the shrinking and the stepwise enlargement of pressure creating its own complex profile [15]. The ejector-type MBG also generates vacuum pressure by implementing the converging-diverging nozzles [23]. The pressure energy of flowing liquid is altered by the velocity change, as such, it creates a low pressure below the atmospheric one to draw in and to entrain the suction gas. Then, turbulence liquid flow induces shear on the entering gas and sweeps it to form microbubbles. The ejector forms microbubbles with the diameters of about 40 to 50 μm. On the other hand, a recent study reported that the diameters of microbubbles formed by the venturi type of MBG are in a range of 100–300 μm [24].

Most of the previous studies focus on examining the underlying mechanism of the microbubble formation and their dynamics. However, only a few studies focus on addressing the operation of the venturi/orifice type MBG, especially with respect to energy input. Therefore, this study addresses these research gaps by investigating the operational parameters of a porous venturi-orifice MBG for oxygen dissolution in water. The study was focused on the effect of liquid velocity and gas velocity ($Q_g$) on the generated vacuum pressure and the oxygen mass transfer rate as well as on the aeration efficiency associated with them. The aeration efficiency parameter is very important to gauge the current state of the MBG technology in comparison with another established oxygenator. The novelty of this study is on the design on MBG itself, as a combined venturi and orifice structure aimed to minimize the energy loss, due to its friction reduction capabilities. Assessment of such MBG system in term of energy efficiency is still not well explored in the literature. Previous reports addressed different types of MBG on their effectiveness for oxygen dissolution, and on the mechanism of microbubble formation and the dynamics of the bubble size and size distribution. On the other hand, this study addresses the knowledge gap on the impact of operational parameters (gas velocity and liquid velocity) towards the rate of oxygen dissolution using the venturi-orifice type MBG. The aim is to understand the behavior of oxygen transport from gas to liquid phase before conducting further operational optimization, or even optimizing the MBG design. The liquid velocity range was set from 2.36 to 2.60 m/s (35–40 L/min), which corresponds to the range which is sensitive to the bubble size (please see reference [24]). It also includes the assessment of the aeration efficiency (kgO$_2$/kWh), which allows us for a better comparison with other MBGs and other established aeration systems.

## 2. Materials and Methods

### 2.1. Materials

The experiments were conducted at room temperature of 20 °C, using tap water as medium for oxygen dissolution. Before each experiment, the initial dissolved oxygen (DO) of the water was measured and 7.9 ppm of sodium sulfite ($Na_2SO_3$, R&M Chemicals, London, UK) per ppm of oxygen was added into the water to for deoxygenation [21,24]. A total of 1.5 ppm of cobalt chloride (Sigma-Aldrich, St. Louis, MO, USA) was added too, as the catalyst for the oxygenation reaction. The initial DO concentrations in the raw water were about 4.0–4.5 ppm. The reaction of deoxygenation is shown below.

$$Na_2SO_3 + \frac{1}{2} O_2 \rightarrow Na_2SO_4$$

### 2.2. Experimental Setup

The setup used in the experiment is illustrated in Figure 1. The experiment was conducted in 1.5 m height cylindrical tank with effective water content of 700 L. The large volume of water was applied to allow slow development of DO during the test and thus can be used to accurately calculate the volumetric mass transfer coefficient ($K_La$). Two probes of DO meters were placed at locations of 0.3 and 1.45 m from the water surface level. The average of the DO concentration readings was taken for the data analysis. A submersible water pump (HJ5500, 100 Watt, Sunsun, AKD5500, Chennai, India)

was placed near the bottom of the tank in which the inlet came from the side and the outlet faced upside (Figure 1). The porous venturi-orifice MBG was placed at the discharge line of the pump.

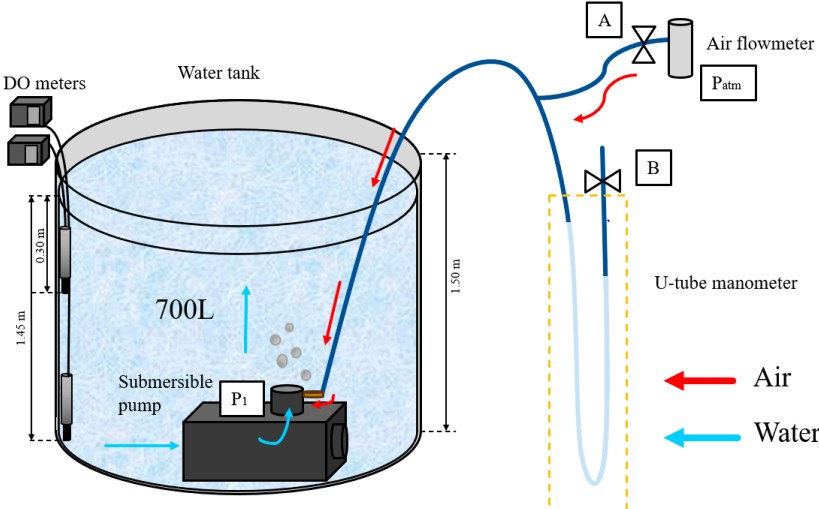

**Figure 1.** Illustration of experimental setup, in which $P_1$ is the vacuum pressure measured using the water manometer, while A and B are valves to adjust the velocities.

The design of a custom-made porous venturi-orifice MBG structure installed in the setup is shown in Figure 2. The MBG was installed on the discharge line atop the submersible pump. The structure consists of a venturi pathway for liquid inlet, and a 10 mm orifice ring. Combination of the venturi and orifice was applied to reduce friction loss and to enhance the pressure drop. The case material of the pipe was polyethylene. The porosity was formed by wounding the case with polypropylene net with an estimated porosity of 0.3. The air suction room had a 7 mm opening, connected with a tube for air flow. The air was sucked through a 7 mm pipe from the open atmosphere. A T-connector was used to link the tube to a water manometer to measure the pressure. An air flow meter incorporated with flow regulator was installed near the entrance of the air.

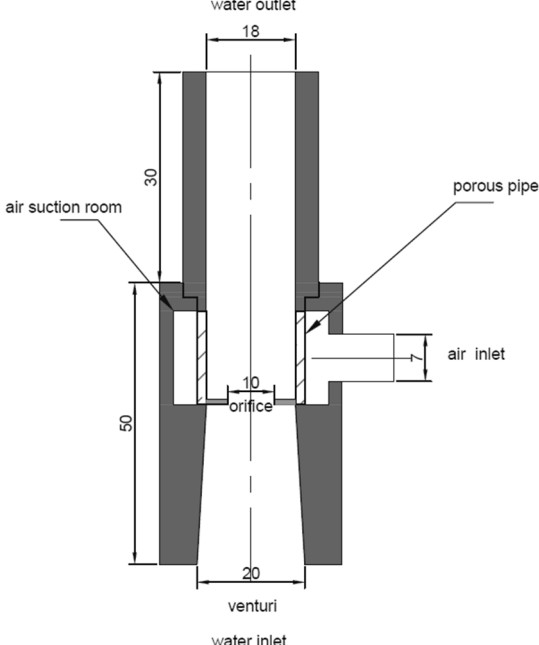

**Figure 2.** Detailed drawing of porous venturi-orifice structure (unit: mm).

## 2.3. Assessment of the System

The venturi-orifice MBG system was first evaluated by assessing the effect of the liquid velocity on the vacuum pressure and the velocity of the free-flowing air. When evaluating the impact of pressure, the air flow was blocked; while when measuring the air velocity, the vacuum pressure was also recorded. This way the pressure drops due to the air flow could also be measured. The tests were run at ten liquid velocities of 0.599, 0.605, 0.612, 0.619, 0.626, 0.633, 0.640, 0.647, 0.654, 0.661 $\times 10^{-3}$ m$^3$/s, corresponding to the linear velocity of 2.35, 2.38, 2.41, 2.43, 2.46, 2.49, 2.52, 2.54, 2.57, 2.60 m/s. The selection of the velocity values was dictated by the ten possible settings of the velocity provided by the applied liquid pump. This test was conducted to explore the operational range of the system as well as to define the range of testing parameters for the oxygen dissolution tests. The liquid velocity was measured at every setting and the obtained data are reported as the liquid velocity values set for the experiments.

## 2.4. Oxygen Dissolution Tests

Before starting each experiment, the DO concentration was lowered to approximately 2 mg/L by dosing appropriate amount of sodium sulfite. There were three types of test conducted for assessment of oxygen dissolution: (1) effect of liquid velocity at constant gas velocity, (2) effect of gas velocity at constant liquid velocity, and (3) effect of liquid velocity at free flow of air. For the first test, the liquid velocity was controlled by switching the knob on submersible pump under the ten settings, corresponding to liquid velocities of 2.35, 2.38, 2.41, 2.43, 2.46, 2.49, 2.52, 2.54, 2.57, 2.60 m/s under a fixed gas velocity of 3 L/min. For the second type, the liquid velocity was fixed at 2.46 m/s, while gas velocities were varied at 1, 2, 3, 4 and 5 L/min. In this case, the air velocity was controlled at the air inlet pipe. For the third test, the air was let to flow freely, in which increasing liquid velocity was accompanied by increasing gas velocity. Each test was performed for one hour and the measurements of DO were taken for every minute. The data of DO concentration against time were used to calculate the $K_L a$. This term is the combination of liquid film coefficient ($K_L$) and interfacial area per unit volume (a). It has linear relationship to the oxygen transfer rate as in Equation (1) [25,26]:

$$\frac{dC}{dt} = K_L a (C^* - C_t) \tag{1}$$

where $C^*$ is the saturation concentration of the DO (mg/L), $C_t$ refers to the concentration of DO (mg/L) at the time (t). Equation (1) can be linearized into Equation (2), which can be used to estimate the $K_L a$. The $K_L a$ is the gradient of a linear plot between $-ln\left(\frac{C^*-C_t}{C^*-C_0}\right)$ vs. time.

$$-ln\left(\frac{C^* - C_t}{C^* - C_0}\right) = K_L a (t - t_0) \tag{2}$$

## 2.5. Aeration Efficiency

The data of DO against time were also used to estimate the aeration efficiency. The aeration efficiency is one of the performance standards for oxygen dissolution devices, including the MBG. The estimation was done for the oxygen transfer for a total concentration increment of 4 mg/L. This value is above the typical required DO concentration in aerobic wastewater treatment of >2 mg/L [27]. Equation (3) was derived from the Bernoulli's equation and the aeration energy was calculated using Equation (4).

$$P = \left(\frac{\Delta P}{\rho} + \frac{v^2}{2}\right) Q_L \rho \tag{3}$$

$$A_E = \frac{V \Delta C}{P \Delta t} \tag{4}$$

where *P* is the pump work (J), $\Delta P$ is the net pressure of the liquid pump (40,000 Pa), $\rho$ is water density (1000 kg/m$^3$), *v* is the linear velocity of water (m/s) and liquid velocity is liquid velocity (m$^3$/s). $A_E$ is

the aeration energy (kgO$_2$/kWh), $V$ is water volume (700 L), $\Delta C$ is the difference in the dissolved concentration within the applied range ($4 \times 10^{-6}$ kg/L) and t is the time to reach $\Delta t$.

## 3. Results and Discussion

### 3.1. Effect of Liquid Velocity on the Vacuum Pressure and Gas Velocity

Figure 3 shows the vacuum pressure created (P$_{atm}$ − P$_{abs}$) with respect to liquid velocity, demonstrating that the applied range of liquid velocity was sufficient to generate vacuum pressure to allow permeation of air through the porous pipe. During the test, the air flow was fully closed and thus no bubble was formed through the MBG. As the pump power was switched on, the water circulated within the set velocities through the MBG. The liquid velocity increased from the inlet pipe to the venturi-orifice tube, hence, causing lower pressure below the P$_{atm}$. It creates a negative pressure (P < P$_{atm}$), which allows the air from the surrounding to be sucked automatically into the MBG. The presence of the porous pipe allowed formation of microbubbles when entering the system. This could be explained by the Bernoulli Principle, in which an increase in fluid velocity within the tube is accompanied by the decrease in the pressure.

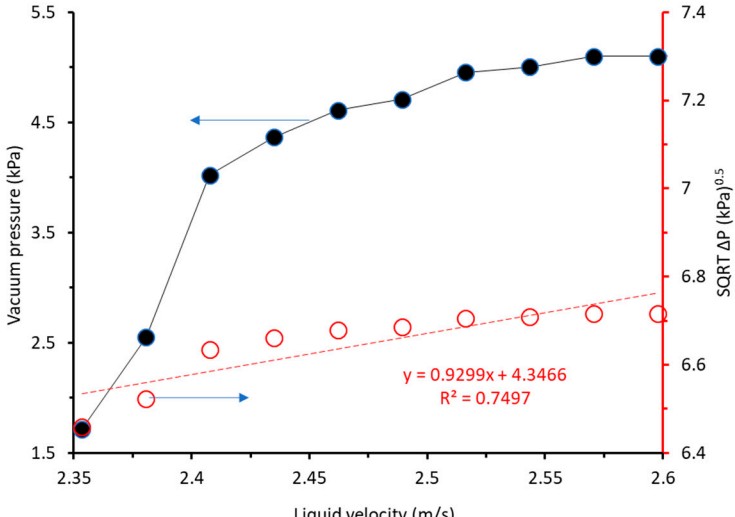

**Figure 3.** Effect of liquid velocity on the generated vacuum pressure in the venturi-orifice microbubble generator (left y-axis) and its relationship with the absolute pressure drop (right y-axis). The tests were run in the absence of air flow through the microbubble generator.

The MBG works based on the Bernoulli Principle, which epitomizes the energy balance principles in which the increase in liquid velocity through the throat leads to lower pressure reaching a vacuum condition. The design of the porous venturi-orifice MBG applied in this study was inspired from the design of Sadatomi and Kawahara type of MBG [6,21], for which no positive pressure was required to force the air which is needed for generating bubbles. The bubble size and distributions were not analyzed in detail, and they will be subjects of future study. However, visual observation on the rising bubbles on top of the tank showed that the bubbles were in millimeter size. The large size of the bubble is expected since the bubbles were depressurized as they rose to the top of the liquid. Analysis of similar type of MBG has been reported earlier but with much lower aeration rates whereby the bubble sizes near the discharge point were around 100–300 µm [24].

Figure 3 also shows that higher liquid velocity leads to higher vacuum pressure (pressure drop). The pressure difference increases sharply with the liquid velocity increment at lower liquid velocity (from 2.35 to 2.41 m/s). However, as the liquid velocity further increases, the increment rate is lower until reaching a condition where the effect of liquid velocity on the pressure is minimum indicated by plateau value of air velocity beyond liquid velocity of 2.5 m/s. The graph of square root function for the

pressure drop against liquid velocity in Figure 3 was derived according to Equation (5). This equation is originally used for calculating dimensionless discharge coefficient ($C_D$) of stream flow in an orifice meter, in which $\beta$ is ratio of orifice diameter to pipe diameter (-), $u_0$ is linear velocity (m/s), $\rho$ density of liquid (kg/m$^3$) and $\Delta P$ is pressure drop (kPa) [28]. The graph can also show that water inlet velocity, $u_0$ (m/s) is linearly proportional to the square root of the pressure drop. This relationship was proven by Shah et al. (2012), using both CFD prediction and experimental data [28]. Nevertheless, it is worth mentioning that the linear relationship as suggested in Equation (5) does not fit really well the experimental data, corresponding to R$^2$ of 0.7497. The deviation from linearity originates largely from the first three data points with liquid velocity of 2.35, 2.38, 2.41 m/s in which prominence impact of the liquid velocity on the vacuum pressure was observed, which requires further detailed analysis.

$$\frac{u_0}{C_D} = \frac{\sqrt{\frac{\Delta P}{\rho}}}{\sqrt{(1 - \beta^4)}} \tag{5}$$

Figure 4 shows the relationship between the liquid velocity and gas velocity. Increasing liquid velocity leads to higher gas velocity, because of the higher vacuum pressure generated inside the porous tube, as shown in Figure 3. The flow of air into the MBG is driven by the vacuum pressure inside the porous tube. When the liquid velocity further increases over 2.46 m/s, its influence on the air velocity is not significant, which also correlates well to the pressure difference pattern presented in Figure 3. Since high liquid velocity corresponds to high pumping energy (as depicted by Equation (3)) but offers small impact on the vacuum pressure and gas velocity, operating an MBG under high cross flow velocity would result in a low aeration efficiency. Therefore, for the further study of the impact of gas flowrate (varying the gas velocity s), the liquid velocity of 2.46 m/s was used as a fixed variable.

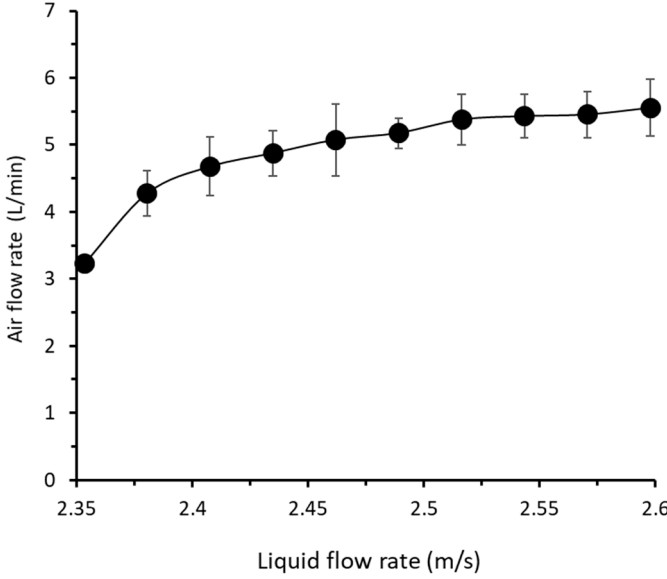

**Figure 4.** Effect of liquid velocity on the air velocity under free-flow of air mode.

Figure 5 shows the relationship between the vacuum pressure generated by the MBG, and the resulting air velocity under free-flowing mode, in which the air was allowed to enter the MBG freely without any restriction. The valve at air flow meter which is connected to the air tube was fully opened. The flow of air was driven by the vacuum pressure and thus was indirectly dictated by the liquid velocity (see Figure 3), where liquid velocity is proportional to the square root of pressure drop and gas velocity. Moreover, it is worth noting that significant increase of the vacuum pressure is observed from the first to the second and the third data point, corresponding to minor increments of liquid velocity

from 2.35 to 2.38 and 2.41 m/s, respectively. However, currently, this finding still cannot be explained, and will be our main subject of the future follow-up study.

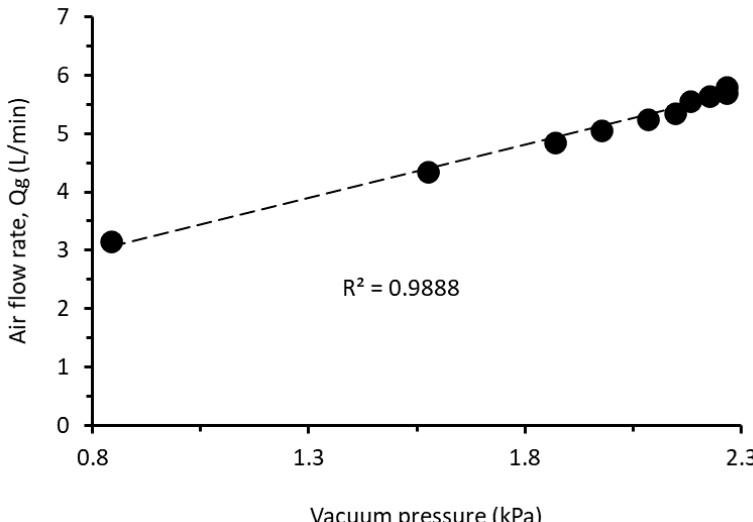

**Figure 5.** A linear relationship of the air velocity to the vacuum pressure generated by the venturi-orifice microbubble generator.

## 3.2. Effect of Liquid and Gas Velocity on the Oxygen Dissolution Rate

Figure 6 shows the profile of DO in water as a function of time and at various liquid velocities. The test was conducted to explore if there is any optimum liquid velocity for oxygen dissolution through microbubbles formation. The DO increment is much higher at the initial stage of the test where the DO concentration is far from the saturation value. The rate of oxygen peaks at the middle range of the tested velocity at 2.43 m/s. Since the air velocity was fixed (3 L/min), the total supply of oxygen to the system was equal for all tests. Therefore, the difference in DO increment rate as a function of time can be attributed to the role of liquid velocity in affecting the mixing and the distribution of bubble sizes. As the velocity increases the sweeping flow of the liquid leads to smaller bubble sizes, which correlates well with previously reported findings [24]. This explains the increase in the DO dissolution rate from the lowest linear velocity value of 2.35 up to 2.46 m/s, beyond which the rate of DO increment decreases. Other reports also pointed out that the range of velocity between 30–40 L/min [19,28,29] plays a significant role in affecting bubble sizes (decrement). The liquid velocity range of 30–40 L/min is exactly the range applied in this study, in which formation of bubble is strongly affected by the shear stress until reaching a point where shear stress has minimum impact. Beyond the liquid velocity of 2.46 m/s, the DO increment decreases, most likely due to over-mixing that promotes intensive bubbles contacts and bubbles coalescence.

The trend of the DO increment pattern can be explained by the formation of bubbles with different diameters as a function of the liquid velocity. Low liquid velocity poses low shear stress from the drag force that sweep the air to form the bubbles. The surface tension force that inhibits the release of the bubbles is constant, therefore increasing the shear stress will lead to the formation of smaller bubbles. In addition, Juwana et al. (2019) reported that this condition ends up causing bubble coalescence around the MBG hence increasing the probability generating bigger bubbles [24]. Formation of the large bubbles decreases the interfacial area which leads to lower oxygen dissolution rate, mainly because of a high applied gas velocity of 3 L/min compared to the one with 1 L/min, which could dampen the effect of liquid velocity on the oxygen dissolution rate. At higher liquid velocities, the inertia force acting on the bubble increases causing the bubbles to have a shorter attachment period with the porous structure, at the same time preventing bubbles from merging together. Thus, the bubble generated is smaller due to lesser growth time and greater driving force to leave the MBG. This eventually increases the total

surface-to-volume ratio of microbubbles, and directly improves on their oxygen dissolubility in the water. Meanwhile, a microbubble with smaller diameter would have characteristics of slow rising speed (based on Stokes' law), having enough time for oxygen gas to dissolve in the water. In this case, 2.43 m/s can be considered the optimum liquid velocity that achieves this target. Detailed analysis on different forces acting on the bubble formation can be obtained elsewhere [29].

Liquid velocity s above 2.43 m/s also shows lower oxygen dissolution rates. In this case, it could be linked with the coalescence of bubbles. The vigorous flow of water causes microbubbles lacking time to flow towards the discharge outlet before colliding and combining with one another. The merging of the bubbles leads to uneven bubbles distribution, which is common but unfavorable for aeration purposes, and most importantly resulting in lower interfacial area for mass transfer.

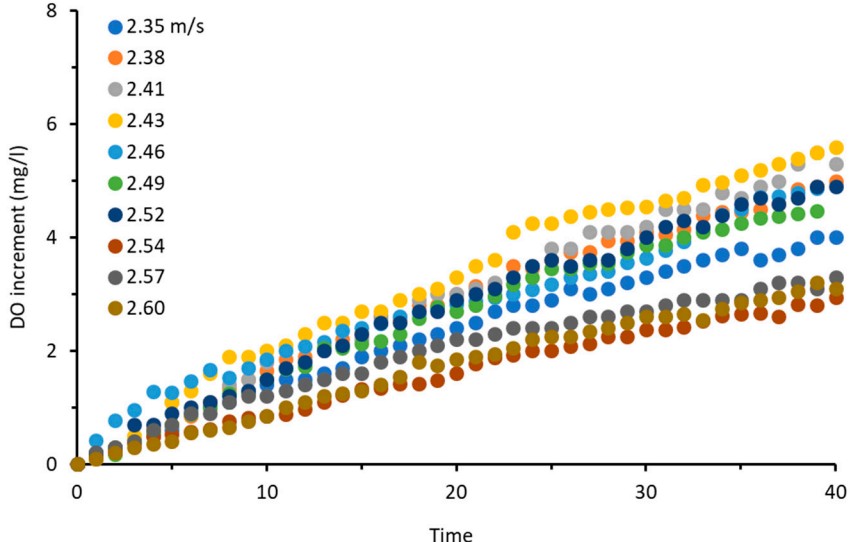

**Figure 6.** Effect of liquid velocity on dissolved oxygen (DO) increment against time at fixed gas velocity of 3 L/min. The DO increment is presented as relative to the initial DO reading.

Figure 7 shows the effect of gas velocity on the oxygen dissolution rate at fixed liquid velocity of 2.43 m/s, in which higher gas velocities lead to higher oxygen dissolution rate. The increment was significant from 1 to 2 L/min, while less so under air velocities of 2–5 L/min. The finding suggests that there is a threshold air velocity that can offer maximum oxygen dissolution rate, which is below 2 L/min. For the aeration rate of 2–5 L/min. The findings can be explained as follows. At low gas velocities and about equal bubbles size, a lower volume of air is available, resulting in lower interfacial area for oxygen mass transfer [12]. This is also due to the possibility of microbubbles being trapped in the porous pipe and, hence, leading to permeability reduction. It seems that, below 2 L/min, the momentum force of the moving air plays an important role in determining the formation and the size of the bubbles. Higher liquid velocity enhances the drag, momentum and pressure forces as detailed elsewhere [29]. Therefore, under very high cross flow velocities all of those forces dictating the air bubbles formation mechanisms, the air velocity does not affect the bubble size and mass transfer area greatly. It was stated that the effect of air velocity is only significant on the bubble size under a range of 0.1–1 L/min [19,30]. According to Al-Ahmady (2005), a greater volume of air supply directly increases oxygen dissolution capacity [25]. This means that total air volume is definitely affecting the oxygen dissolution rate, despite the fact that a smaller microbubble has a greater dissolubility rate. Sadatomi et al. (2012) stated that, when gas velocity of <10 L/min, the oxygen absorption efficiency is roughly independent of gas velocity and of the type of MBG employed [21]. Since this study falls under this gas velocity range (<10 L/min), the conclusion is similar with the minor DO increment for gas velocities of 2–5 L/min. Within this range of gas velocity, the increase in gas velocity leads to production or larger bubble size resulting in slight increment in the oxygen dissolution rate. This finding suggests that overflowing of

air bubbles into the system might not necessarily lead to an effective dissolution process if the bubble size is too large (poor interfacial mass-transfer area). It also means that the system can operate at relatively low crossflow velocity leading to lower energy input. Nevertheless, rigorous analysis of the aeration efficiency must be performed to decide the most optimum operational condition.

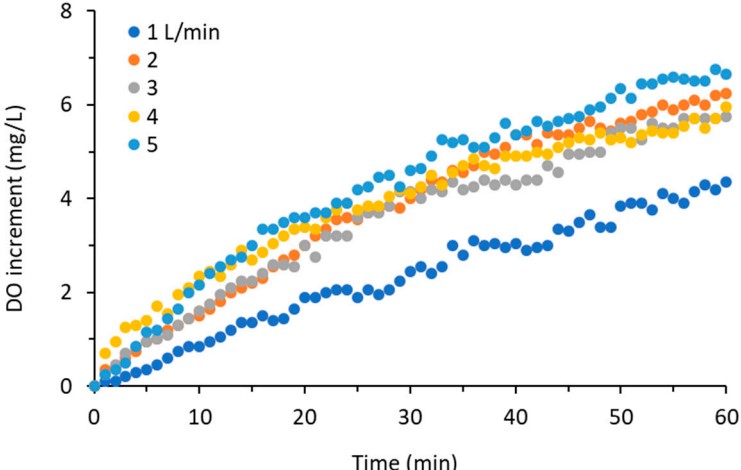

**Figure 7.** Effect of air velocity on dissolved oxygen (DO) increment against time at fixed liquid velocity of 2.46 m/s. The DO increment is presented as relative to the initial DO reading of about 2 mg/L.

Figure 8 shows the DO increment under the free-flowing air condition, where the higher crossflow velocity leads to greater oxygen dissolution rate. Since there is no restriction in the air tube, the air velocities were maximum with respect to each liquid velocity shown in Figure 4. It shows a clear trend in which increasing the liquid velocity leads to higher oxygen dissolution rate. This can be explained as a higher air velocity leads to a higher volume of the air being introduced into the system coupled with the formation of about similar sized bubbles (Figure 8). In this condition, the oxygen dissolution rate seems to correlate well with the air/liquid interface, which promotes the mass transfer of oxygen.

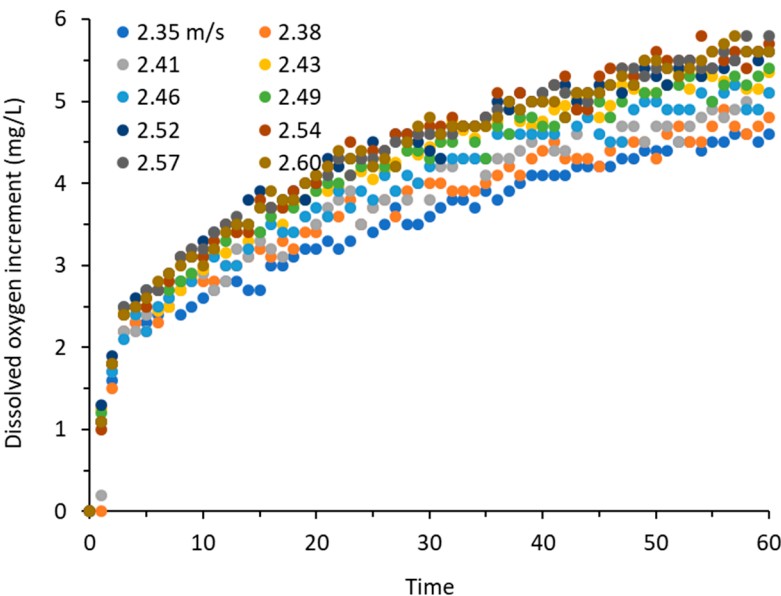

**Figure 8.** Dissolved oxygen (DO) increment against time at free flow of gas velocity. The DO increment is presented as relative to the initial DO reading.

Interestingly, the oxygen dissolution rates under maximum air flow (Figure 8) are lower than the one with restricted air velocity (Figure 6). Referring to Figure 7, overflowing of air bubbles does not guarantee a greater oxygenation rate. This demonstrates the importance of bubble size in affecting the oxygen dissolution rate. Despite of lower rate of air flow, the high rate of oxygen dissolution is enhanced by formation of smaller bubbles leading to higher gas/liquid interfacial area. This finding suggests the necessity for operational optimization of the venturi-orifice type of MBG, to yield maximum dissolution rates. Simply letting free-flowing air with maximum velocity does not lead to a maximum oxygen dissolution rate. It is worth noting that the ranges of bubble size formed for the tests reported in Figures 6 and 8 seem to be significantly different, judging from the rates of oxygen dissolution. As reported earlier, for similar venturi-orifice MBG system, operated at 30–40 L/min, the resulting bubbles sizes were in range of 450–1000 μm when the air velocities were set to 0.1–1 L/min [24]. However, since no measurement of bubble size was conducted, this merely remains just a conjecture.

### 3.3. Relationship of Liquid and Gas Velocity with Volumetric Mass Transfer Coefficient

The impact of air velocity on the oxygen dissolution rate is not conclusive, and somewhat counterintuitive. To understand the behavior of the oxygen transfer, it can be further analyzed using the $K_La$ values as reported in Figure 9. The $K_La$ counts the impact of bubble velocity (the contact time of bubble with the liquid), bubble diameter (the gas/liquid interfacial area or effective mass transfer area), dynamic viscosity of the liquid (mixing) and the mass diffusivity [31], in which the first two are seen as having the most prominence in this study.

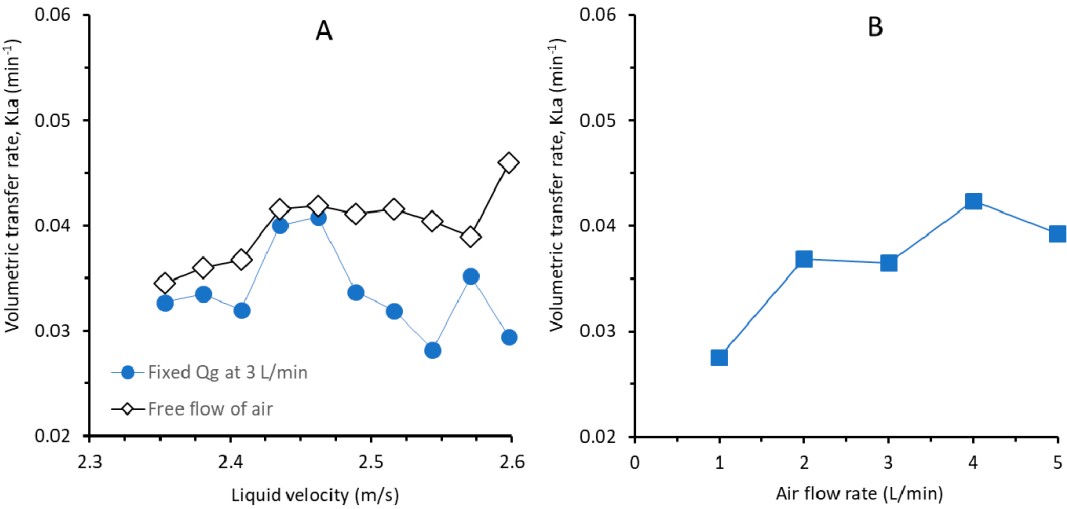

**Figure 9.** Effect of liquid velocity at fixed air velocity of 3 L/min and under free flowing of air (**A**) and the effect of air velocity at a constant liquid velocity of 2.46 m/s (**B**) on the volumetric mass transfer coefficient.

The impact of liquid velocity on the $K_La$ under the free-flowing air shows an increasing trend from 2.35 up to 2.46 m/s (Figure 9A), after which the $K_La$ decreases slightly until the liquid velocity reaches 2.57 m/s. The $K_La$ then suddenly jumps at the highest liquid velocity of 2.6 m/s. The steady increase of the $K_La$ can be ascribed by the increasing air velocity that forms higher number of bubbles hence higher interfacial area for oxygen mass transfer. For the liquid velocities beyond 2.46 m/s, both the liquid and air flows promote bubbles coalescence which eventually reduces the effective mass transfer area. The spike of the $K_La$ for the liquid velocity of 2.6 m/s is presumably due to the smaller bubble sizes produced.

Figure 9B shows that increasing air velocity at constant liquid velocity leads to higher $K_La$. The finding suggests that higher liquid velocity leads to increasing number of bubbles that eventually enhances the area for oxygen mass transfer. The significant increment of air velocity from 1 to 2 L/min suggests that the pressure and momentum forces dictate the formation of the bubbles. For air velocities higher than 2 L/min, the increment is less significant indicating that additional volume of air form slightly larger bubbles only modestly affects the overall effective mass transfer area. It is worth noting that the $K_La$ value is system specific, and the value is affected by the applied experimental set-up. The $K_La$ values obtained in this study cannot be compared with the ones in reference. Nonetheless, the trend of $K_La$ obtained in this study is in line with an earlier report [24]. The $K_La$ increasing as the air velocity increases under free air flow system and the increasing trend of $K_La$ as function of air velocity at constant liquid velocity have been also reported elsewhere [24].

### 3.4. Aeration Effciency

Since the $K_La$ value is system specific and not directly comparable within data obtained from different experimental setups, a universal parameter in form of the specific aeration efficiency is used to assess the system (presented in Figure 10). The trend of the aeration efficiency is similar to the $K_La$. The energy efficiency for operation of the venturi-orifice system peaks at value of 0.424 kgO$_2$/kWh under the free-flowing air at a liquid velocity of 2.54 m/s, corresponds to the $K_La$ value of 0.0404 min$^{-1}$.

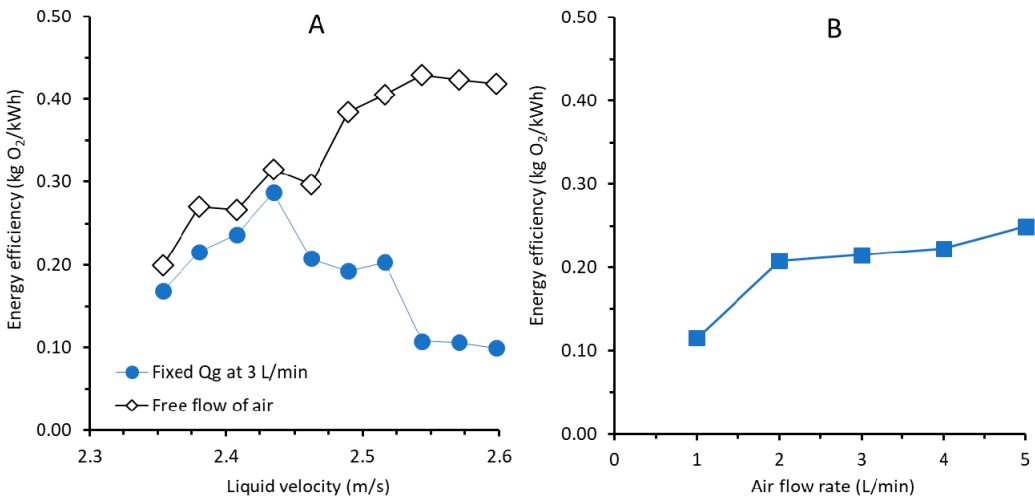

**Figure 10.** The effect of liquid velocity at constant air velocity of 3 L/min and free flow systems (**A**) and the effect of air velocity at fixed linear velocity of 2.46 m/s (**B**) on the aeration efficiency.

The maximum aeration efficiency value obtained in this study is greatly below the established aerators which are used in large scale industries. The typical energy efficiencies of surface aeration and fine bubble aeration systems for dissolution of oxygen from air into clean water are 1.1–2.0 and 2.0–5.5 kgO$_2$/kWh and for dissolution of oxygen from air into wastewater are 0.9–1.7 and 1.3–2.6, respectively [32]. The identical trend on the energy efficiency to the $K_La$ suggests that it is strongly affected by the size of the formed bubbles. Nonetheless, it is worth noting that the venturi-orifice MBG system tested in this study is still not optimized yet and could be improved further to enhance its energy efficiencies.

Section 3.2 and 3.3 discuss the impact of operational parameters on the oxygen transfer rate and $K_La$. The findings unravel the importance of optimizing operational parameters in obtaining the highest $K_La$ (small bubbles). Most of the previous study on MBG put emphasis on the mechanics of microbubble formation and the bubble size dynamics [27,29,30,33]. Such information should be used as input for designing an energy efficient oxygenator that improves the performance of the current established systems. By referring to earlier report, a very low ratio of gas and liquid velocity

needs to be implemented (<0.033) to form micron size bubbles [24]. It means that a high pumping energy is required to create small volumes of air bubble by applying high liquid crossflow velocity. The formation of microbubbles offers a maximum effective surface area and a longer retention time in the water for prolonged mass transfer to occur. This way the dissolved oxygen can also be enhanced. However, since the ratio of gas to liquid velocity is too small, they only carry limited amount of oxygen in the air, which becomes the limiting factor for energy efficiencies.

In order to supply ample amount of oxygen, it is proposed that multiple MBGs are required, resulting in inflated energy input to the system. Vice versa, the formation of large air bubbles reduces the specific mass transfer area. Despite of the over-supply of oxygen at higher air velocity but at large bubble sizes, the oxygen dissolution yield remains low due to fast bubble rising velocity that shorten contact of bubbles/liquid. Such trade-off situation necessitates process optimization, as well as a redesign of the venturi-orifice MBG that will offer high oxygen dissolution energy efficiency. Another option is to develop porous tube materials such as hydrophobic membrane to allow formation of smaller air bubbles [34].

## 4. Conclusions

The performance of a porous venturi-orifice MBG was evaluated. The range of operational parameters enables the system to operate under vacuum and increased liquid velocities (from 2.35 to 2.60 m/s) result in higher vacuum pressure (of 0.84 to 2.27 kPa). They correspond to air velocities for the free-flowing air of 3.2–5.6 L/min. For operations under a constant air velocity of 3 L/min but variable liquid velocities, the $K_L a$ peaks at 0.041 min$^{-1}$ corresponding to liquid velocity of 2.46 m/s, which corresponds to a aeration efficiency of 0.287 kgO$_2$/kWh. Only slight increments were achieved on both $K_L a$ and aeration efficiency when the system was operated under variable liquid velocities and under free-flowing air, with the maximum aeration efficiency reaching the maximum at 0.424 kgO$_2$/kWh. The value unfortunately still falls far below the established aerators that are used in large scale industries. The analysis on the energy efficiency revealed that the venturi-orifice MBG could be further optimized by focusing on the trade-off between air bubble size and the air volume velocity in order to establish a greater balance between the amount of available oxygen (to be transferred) and the rate of the oxygen transfer.

**Author Contributions:** K.C.S.L. performed all experiments, data analysis and prepared the manuscript draft. A.R. validated the experimental data and assembled the experimental setup. W.B. designed the setup and revised the manuscript. M.H.M.Y. performed data analysis and revised the manuscript draft. N.S. revised the manuscript draft and performed critical analysis of the data. M.R.B., N.A.H.M.N. and Z.A.P. formulated the conceptual design of the experiments, supervised the laboratory works and edited the manuscript draft. All authors have read and agreed to the published version of the manuscript.

**Funding:** Kelly Chung Shi Liew is supported by Graduate Assistantship scholarship center of post-graduate study Universiti Teknologi Petronas. The APC is supported by Short-term Internal Research Funding of Universiti Teknologi PETRONAS (Grant number: 015LA0-016).

**Acknowledgments:** We acknowledge chemical engineering department Universiti Teknologi Petronas for access to teaching laboratory to conduct the experiments.

**Conflicts of Interest:** The authors declare no conflict of interest.

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
