# Peer review of "Porous Venturi-Orifice Microbubble Generator for Oxygen Dissolution in Water"

_processes, doi:10.3390/pr8101266_

Round 1
Reviewer 1 Report
The article addresses the dissolution of oxygen in water using a microbubble generator in the shape of a venturi tube. The topic is of interest in the field of wastewater treatment, especially in terms of the biological stage of water purification, where aeration is needed.
The article has a medium complexity following the degree of oxygen dissolution in different operating conditions: constant gas flow (oxygen) and variable liquid flow, constant liquid flow and variable oxygen flow, constant liquid flow and free flow of oxygen.
The way the experiments are performed is clearly described, but the presentation of the results could be improved, namely:
-in figure 3 the meaning of the black and red symbols should be described, either inside the figure or in the title of the figure (depending on the requirements of the magazine), because as the figure now shows the meaning of the red symbols is not clear ; In addition, with regard to the regression of the data designated by the red symbols, it can be seen very clearly that they do not fit at all on the equation of a line and for this reason I consider that another form of the equation (of a curves for example) for the regression of the data designated by the red symbols;
-for equation 5 the meaning of each term involved in the relationship should be described;
The observations and conclusions are strictly valid for the operating conditions chosen for the type of bubble generator chosen and cannot be extended to other operating conditions.
There are also some comments related to the drafting:
- page 3, lines 101-103, must be reformulated, the sentence is incomplete.
-page 3, line 119, replace "in at room" with "at room"
-page 13, line 423, appears twice "of the" after "despite".
Reviewer 2 Report
The study describes oxygen mass transfer in water resulting from the operation of a custom-made Venturi-type microbubble generator. The writing style is confusing and the paper is very difficult to read. The presentation should be significantly improved. The novelty and significance of the research with respect to other works, such as ref. [17], which is a much more detailed and coherent investigation, is not clearly presented. I don’t see how the results presented in this work could be useful to other researchers. I don’t recommend publishing the paper in its current form.
There are also some minor points:
Line 91 – what is “energy conversion”?
Line 92 – what is “pressure energy”?
Check grammar/spelling. Overall OK, but some parts are difficult to understand
Line 120 – what is “DO”? All abbreviations must be explained on first use
What is the material of the porous pipe in Fig. 2? What is the porosity and size of the pores?
What is the initial oxygen concentration in the water before deoxygenation?
At the end of section 2.3 it is said that the flow circulation velocity is determined by pump setting. Have these values been confirmed by independent measurement?
Line 174 repeats the information in Line 162
Figure 3 is incomprehensible. What are black symbols? What are red symbols? The Figure (including caption) should be self-contained and comprehensible. Line 224 – “Figure 3 also shows that higher QL leads to higher vacuum pressure (pressure drop)” – Figure 3 does not show this, because “QL” is nowhere to be found in Figure 3.
What is \beta in Eq. (5)?
In Fig. 6 the DO concentration dynamics at QL=2.43 m/s seems to be an outlier. It is not clear how reproducible this effect is and the dramatic difference from the other measurements is not clearly explained.
Reviewer 3 Report
General comment:
In principle the objective of the study is interesting and relevant (investigation and evaluation of the energy efficiency of a porous venturi-orifice microbubble generator). The energy for aeration is the greatest proportion of the whole energy demand in an activated sludge process for wastewater treatment. The generation of microbubbles with the lowest possible diameter enhances the gas/liquid mass transfer and could result in an improved energy efficiency and lower greenhouse gas emissions for energy production.
The following issues should be discussed:
- Page 3: The reaction with sodium sulphite for deoxygenation should be described. In which way does cobalt chloride act as catalyst (for which reaction)?
- Page 7, equation (5): The meaning of the parameter “β” has to be explained.
- Is it possible to estimate the savings potential concerning energy and greenhouse gas emissions if the investigated MBG would be used instead of a conventional aerator?
- Are there any practical benefits of the results besides the scientific findings?
Furthermore, there are a number of errors that should be corrected. The text contains a number of mistakes and incomprehensible sentences/phrases. So in the final analysis, a major revision of the entire manuscript is needed.
The specific comments are summarized in the attached pdf file “Specific comments_processes-903234”.

Reviewer 4 Report
General review:
The manuscript presents a systematic an analysis of oxygen dissolution during hydrodinamic cavitaiton with some interesting conclusions regarding its effect on bubble sizes and oxygen transfer. Even though I made an effort to correct the grammatical limitations of this script, some small parts of the text still need to be grammatically corrected.
Specific review:
L25 Better to write: The increment of stream velocity along the venturi pathway and orifice ring leads to pressure drop (Patm>Pabs) and subsequently to increased cavitation.
68 When liquid enters the throat at a higher velocity, it creates lower static pressure and it can be used for air sucking to form microbubbles when the pressure is below the atmospheric ADD REFERENCES HERE: [15] (Franc 2006; Kosel et al. 2017).
Franc J-P. Physics and Control of Cavitation. 2006.
Kosel J, Gutiérrez-Aguirre I, Rački N et al. Efficient inactivation of MS-2 virus in water by hydrodynamic cavitation. Water Res 2017;124:465–71.
47 Better to write: In intensive aquaculture of tilapia fish, application of MGB, as an aerator, also promoted the growth rate of fish (both thier length and weight) [5].
54 Better to write: Some of the recent developments include a system based on a porous media, constant flow nozzles and membrane or gas spargers coupled with a mixer (i.e., impeller) [8].
64 Better to write: Then, the gas-liquid mixture is reduced into microbubbles due to the shear effect of centrifugation which is formed by the rapid rotating liquid flow [11, 14].
66 Better to write: The venturi effect has also been exploited to generate microbubbles and the factors affecting microbubble formation have widely been discussed.
68 Better to write: When liquid enters the throat at a greater velocity, it lowers the static pressure and this effect can be used for air suction and the subsequent formation of microbubbles (static pressure falls below the atmospheric pressure) [15].
70 Better to write: Orifice type MBGs work under similar principles and for the venturi type MBGs the velocity change is also used as a decompressor [16, 17].
75 Better to write: The low local pressure within the venturi tube promote cavitaiton generation conditions, but soon the formed void collapses and the pressure is recovered further downstream [18].
88 Better to write: According to Terasaka et al. (2011), a typical ejector type MBG refers to a liquid flow channel that involves the shrinking and the stepwise enlargement of pressure creating its own complex profile [11].
90 Please clarify: The ejector is also applying the venturi effect of converging-diverging nozzles for energy conversion despite using complex shape of the flow pathway [20].
92 Better to write: The pressure energy of flowing liquid is altered by the velocity change, as such it creates a low pressure below the atmospheric one to draw in and to entrain the suction gas [20].
95 Better to write: On the other hand, a recent study reported that the diameters of microbubbles formed by the venturi type of MBG are in range of 100 – 96 300 μm [17].
98 Better to write: Most of the previous studies focus on examining the underlying mechanism of the microbubble formation and their dynamics.
99 Better to write: However, onyl a few studies focus on addressing the operation of the venturi/orifice type MBG, especially with respect to energy input.
100 Better to write: Therefore, this study addresses these research gaps by investigating the operation of a porous venturi-orifice MBG for oxygen dissolution in water.
101 Please clarify this sentance: is the experiments were focused on the effect of QL and Qg on the generated vacuum pressure and the 102 mass transfer rate as well as the energy efficiency associated with it.
103 Please clarify this sentance: This aspect of study is very important to gauge the current state of the MBG technology in comparison another established 104 oxygenator.
105 Better to write: The novelty of this study is the design of the MBG itself, as a combined venturi and orifice structure which minimizes energy consumption due to its friction reducion cababilities.
109 Better to write: This study addresses the knowladge gap on the impact of operational parameters (Qg and QL) towards the rate of oxygen dissolution within the venturi-orifice type MBG.
113 Better to write: The QL range was set from 2.36 to 2.60 m/s (35 – 40 L/min), which corresponds to the range which is sensitive to the bubble size (please see [17]).
114 Better to write: It also includes the assessment of the aeration efficiency (kgO2/kWh) which allows us for a better comparison with other MBGs and other established aeration systems.
204 Please clarify and make better sentances: This created a negative pressure (P<Patm) which allows air from the surrounding being sucked automatically into the MBG without other driving force that consumed more energy. While the mesh-like porous design enable smaller size microbubbles formation when the air form bubble in the gas/liquid system.
211 Better to write: The design of the porous venturi-orifice was inspired from the design of Sadatomi and Kawahara type of MBG [6], [16]], for which no positive pressure was required to force the air which is needed for generating bubbles.
213 Please rephrase: Detailed analysis on the size of the bubble was not performed in this study because it is considered as beyond the scope.
225 Please clarify this sentance: However, as the QL further increases, the increment is lower until reaching a condition where the effect of QL on the pressure is minimum.
232 Please clarify this sentance: Nevertheless, it is worth to mention that the linear relationship does not fit really well on the first three data points as showing a much more prominence impact of the QL on the vacuum pressure, which require further detailed analysis.
240 Please clarify this sentances: As higher energy input is consumed at higher velocity, only small impact is obtained on the vacuum pressure and gas flow rate.
242 Better to write: Therefore, for the further study of the impact of gas flowrate (varying the gas flow rates), the liquid flow rate of 2.46 m/s was used as a fixed variable.
250 Better to write: The flow of air was driven by the vacuum pressure and thus was indirectly dictated by the liquid flow rate (see Figure 3), where QL is proportional to the square root of pressure drop and Qg.
254 Better to write: However, currently this finding still cannot be explained and will be our main subject of the future follow-up study.
260 Better to write: Figure 6 shows the profile of DO in water as a function of time and at various liquid flow rates.
265 Better to write: Therefore, the difference in DO increment rate as a function of time can be attributed to the role of liquid velocity in affecting the mixing and the distribution of bubble sizes.
267 Better to write: As the velocity increases the sweeping flow of the liquid leads to smaller bubble sizes wich corelates well with previously reported findings [17].
270 Better to write: Other reports also pointed out that the range of flow rate between 30 – 40 L/min [17, 30, 31] plays a significant role in affecting bubble sizes (decrement).
277 Please clarify this sentances: At low liquid flow rates, it poses lower shear stress in the form of the drag force that sweep the air to form the bubble.
278 Better to write: The surface tension force that inhibits the release of the bubbles is constant, therefore increasing the shear stress will lead to the formation of smaller bubbles.
280 Better to write: In addition, Juwana et al (2019) reported that this condition ends up causing bubble coalescence around the MBG hence increasing the probability of generating bigger bubbles [17]. Formation of the large bubbles decreases the interfacial area which leads to a lower oxygen dissolution rate.
283 Please clarify this sentance: This explanation is acceptable, especially since a high gas flow rate was applied in this study (3 L/min) compared to the one with 1 L/min [17], which could have less obvious impact of liquid velocity on the oxygen dissolution rate.
285 Better to write: At a higher liquid velocities, the inertia force acting on the bubble increases causing the bubbles to have a shorter attachment period with the porous structure, and at the same time preventing bubbles to merge togheter.
295 Better to write: However, liquid flow rate above 2.43 m/s also shows lower oxygen dissolution rates.
308 Please clarify this sentance: At a lower gas flow rate, limited volume of air can be used for microbubble formation, hence when the bubble sizes do not differ significantly, it reduces oxygen dissolution rate as also suggested elsewhere [8].
312 Better to write: It seems that below 2 L/min, the momentum force of the moving air plays an important role in determining the formation and the size of the bubbles.
314 Please clarify this sentance: Since the drag force is constant at constant liquid velocity, increasing air flow rate corresponds well with increasing the momentum and pressure forces [26], in which increasing the air flow rate enhances mass transfer interface from formation of more bubbles with minimum impact on the bubble size. This is proven as bubble generation efficiency is rising along the air supply even though being reported that a lower air flow rate contributes to smaller microbubbles under 0.1 – 1 L/min [17, 32].
319 Better to write: According to Al-Ahmady (2005), greater volume of air supply directly increases oxygen dissolution capacity [22]. This means that total air volume is definitely affecting the oxygen dissolution rate despite the fact that a smaller microbubble has a greater dissolubility rate.
322 Better to write: Sadatomi et al. (2012) stated that when Qg < 10 L/min, the oxygen absorption efficiency is roughly independent of Qg and of the type of MBG employed [16].
325 Please clarify this sentance: Within this range of Qg, the impact of microbubbles size and total volume seems to be balancing within each other thus resulting in slight increment in the oxygen dissolution rate.
326 Better to write: This finding suggests that overflowing of air bubbles into the system might not necessarily lead to an effective dissolution process if the bubble size is too large (poor interfacial mass-transfer area).
339 Better to write: This can be explained because a higher air flow rate leads to a higher volume of the air being introduced into the system coupled with the formation of about similar sized bubbles (Figure 8).
348 Better to write: This finding suggests the necessity for operational optimization for the venturi-orifice type of MBG to yield maximum dissolution rates.
349 Better to write: Simply letting free-flowing air with maximum flow rate does not lead to a maximum oxygen dissolution rate.
354 Better to write: However, since no measurement of bubble sizes was conducted, this merely remains just a conjecture.
355 Exclude this sentance: To include the impact of bubble surface area on the oxygen dissolution rate, it is discussed in terms of volumetric mass transfer coefficients in Section 3.3.
374 Please clarify this sentance: The steady increase of the KLa can be explained by increasing the volumetric rate of the air that for higher number of bubbles.
377 Better to write: The spike of the KLa for the liquid velocity of 2.6 m/s is presumably due to the smaller bubble sizes produced.
379 Please clarify this sentance: It is presumed that an increasing number of bubbles with higher liquid velocity hence the effective mass transfer area of oxygen for mass transfer.
382 Better to write: For the air flow rates higher than 2 L/min, the increment is less significant indicating that additional volume of air form slightly larger bubbles only modestly affect the overall effective mass transfer area.
385 Please clarify this sentance: Comparison of the data obtained in this study under different parameters is valid, which is used to understand the behavior of the oxygen transport from gas to liquid phase. However, the KLa value cannot be compared with the one obtained from different setup.
393 Better to write: Since the KLa value is system specific and not directly comparable within different experiments, a universal parameter of specific aeration efficiency is used to assess the system (presented in Figure 10). The trend of the aeration efficiency is similar to the KLa trend. The energy efficiency for operation of the venturi-orifice system peaks at value of 0.23 kgO2/kWh under a fixed air flow rate of 3 L/min and at a liquid velocity of 2.43 m/s.
401 Better to write: The maximum aeration efficiency value obtained in this study is greatly below the established aerators which are used in large scale industries.
422 Better to write: Vise versa, formation of large air bubbles reduces the specific mass transfer area.
423 Please clarify this sentance: Despite of the of the over-supply of oxygen, the oxygen dissolution yields is low due to fast bubble rising velocity that shorten contact of bubbles/liquid.
425 Better to write: Such trade-off situation necessitates for process optimization as well as for a redesign of the venturi-orifice MBG that will offer a higher oxygen dissolution energy efficiency.
428 Better to write: The performance of a porous venturi-orifice MBG was evaluated. The ranges of possible operational parameters enable the system to operate under vacuum and increased liquid velocities (from 2.35 to 2.60 429 m/s) result in higher vacuum pressures (of 0.84 to 2.27 kPa).
431 Better to write: The tests on the dissolution rate reveals that under a constant air flow rate of 3 L/min (at various liquid velocities) the rate of oxygen dissolution peaks at liquid velocity of 2.43 m/s which also provides the highest KLa (0.043 min-1) and the most energy efficient condition (0.23 kgO2/kWh).
436 Better to write: The analysis of the energy efficiency of the venturi-orifice MBG revealed that its efficiency could be further optimized by focusing on the trade-off between air bubble size and the air volume flow rate in order to establish a greater balance between the amount of available oxygen (to be transferred) and the rate of the oxygen transfer.
Round 2
Reviewer 2 Report
The quality of the manuscript has improved after revision. Some of my concerns have been addressed. It can be published in the present form.
Author Response
Reviewer Comment
The quality of the manuscript has improved after revision. Some of my concerns have been addressed. It can be published in the present form.
Response to the reviewer comment
Dear Reviewer,
Thank you for the positive comments. We feel grateful of getting quality feedback from reviewer to help in enhancing manuscript quality. We have perform language editing to enhance the readability of the revised manuscript.
Reviewer 3 Report
General comment:
From my point of view the authors have answered most of the questions sufficiently and have also taken the comments of the reviewers into consideration in most cases. Overall, this led to an improvement of the article. In the final analysis, the reviewed article seems to be suitable for publication.
There are only some minor errors that should be corrected:
Page 1, line 25: “...treatment systems...” instead of “...treatment system...”.
Page 1, line 26: “...of the microbubbles...” instead of “...of the microbubble...”.
Page 1, line 27: “...emphasis...” instead of “...emphases...”.
Page 1, line 38: Unknown unit in “0.0416 in-1”
Page 2, lines 97/98: Uniform spelling! “ejector type” or “ejector-type” in all cases.
Page 3, line 104: “...in a range...” instead of “...in range...”.
Page 3, line 123: “...(please see [22]).” instead of “...(please see [22].”.
Page 3, line 123: “...(kgO2/kWh)...” instead of “...(kgO2/kWh)...”.
Page 3, line 132: “...4.0 – 4.5 ppm...” instead of “...4.0-4.5 ppm...”.
Page 7, line 248: “...which requires...” instead of “...which require...”.
Page 8, line 283: “...which corellates...” instead of “...which corelates...”.
Page 9, line 330: “It was stated that the effect of air flow rate only significant on the bubble size...” seems to be incomplete and is incomprehensible.
Page 12, line 392: “...the area for oxygen mass transfer.” instead of “...the area of for oxygen mass transfer.”.
Page 12, lines 411/412: “...below the established aerators...” instead of “...below the established aerator...”.
